# Establishment and Application of a Predictive Growth Kinetic Model of *Salmonella* with the Appearance of Two Other Dominant Background Bacteria in Fresh Pork

**DOI:** 10.3390/molecules27227673

**Published:** 2022-11-08

**Authors:** Ge Zhao, Tengteng Yang, Huimin Cheng, Lin Wang, Yunzhe Liu, Yubin Gao, Jianmei Zhao, Na Liu, Xiumei Huang, Junhui Liu, Xiyue Zhang, Ying Xu, Jun Wang, Junwei Wang

**Affiliations:** 1Laboratory of Pathogenic Microorganisms Inspection, Livestock and Poultry Products Quality & Safety Risk Assessment Laboratory (Qingdao) of MARA, China Animal Health and Epidemiology Center, Qingdao 266032, China; 2College of Food Science and Engineering, Ocean University of China, Qingdao 266003, China; 3College of Food Science and Engineering, Qingdao Agricultural University, Qingdao 266109, China

**Keywords:** pork, *Salmonella*, background bacteria, predictive growth model, shelf life

## Abstract

To better guide microbial risk management and control, growth kinetic models of *Salmonella* with the coexistence of two other dominant background bacteria in pork were constructed. Sterilized pork cutlets were inoculated with a cocktail of *Salmonella* Derby (*S.* Derby), *Pseudomonas aeruginosa* (*P. aeruginosa*), and Escherichia coli (*E. coli*), and incubated at various temperatures (4–37 °C). The predictive growth models were developed based on the observed growth data. By comparing *R*^2^ of primary models, Baranyi models were preferred to fit the growth curves of *S.* Derby and *P. aeruginosa*, while the Huang model was preferred for *E. coli* (all *R*^2^ ≥ 0.997). The secondary Ratkowsky square root model can well describe the relationship between temperature and *μ_max_* (all *R*^2^ ≥ 0.97) or *Lag* (all *R*^2^ ≥ 0.98). Growth models were validated by the actual test values, with *B_f_* and *A_f_* close to 1, and *MSE* around 0.001. The time for *S.* Derby to reach a pathogenic dose (10^5^ CFU/g) at each temperature in pork was predicted accordingly and found to be earlier than the time when the pork began to be judged nearly fresh according to the sensory indicators. Therefore, the predictive microbiology model can be applied to more accurately predict the shelf life of pork to secure its quality and safety.

## 1. Introduction

Fresh pork is currently the most important meat product consumed by Chinese residents, which is rich in nutrients, but also a good medium for microbial growth during production and transportation. The continuous growth of microorganisms at a suitable temperature will eventually lead to meat spoilage or even foodborne disease [1]. *Salmonella*, as the main foodborne pathogen in pork [2,3], may cause infection due to a short cooking time and cross-contamination during treatment in the kitchen. Therefore, it is very important to control microbial contamination and continuous reproduction in pork to ensure the quality and safety of products.

The behavior of microorganisms is an important factor affecting the quality and safety of meat, and temperature is the most important factor that allows for microorganisms to continue to grow and even secrete harmful substances [1]. Traditional microbial detection with certain lag results is time-consuming and labor-intensive, and it cannot predict subsequent bacterial contamination [4]. On the other hand, predictive microbiology is computer-based, describing the growth, survival, and death of microorganisms under specific environmental conditions, and establishing mathematical models to provide a basis for risk assessment [5,6,7,8]. Using it can quickly predict the changes in pathogenic or spoilage bacteria in food according to the primary contaminated quantity without the traditional microbial detection, which realizes intelligent monitoring from raw materials to production to contribute to microbial risk management in food [9,10]. Therefore, predictive microbiology studies on pork are important.

In regard to the predictive microbiological model in pork, previous domestic studies mostly established predictive models for single spoilage bacteria or pathogenic bacteria [11,12,13]. However, in pork, there are often several types of dominant bacteria that contaminate a certain quantity and continue to proliferate at the same time. The specific spoilage bacteria, *Pseudomonas*, are the dominant background bacteria in fresh pork [14], and *Escherichia coli* (*E. coli*) is also a type of major background bacterium [15,16]. During the post-production process, there will be a certain competitive effect among these bacteria to influence each other’s growth. Recently, some microbiological models related to more than one bacterium have been reported, for example, *Pseudomonas* and *Listeria* in beef [17], *Salmonella* and background flora in chicken [8], and Listeria and natural flora in minced tuna [18]. Nevertheless, in pork, there are no reported studies on competitive predictive microbial models of *Salmonella* with the appearance of dominant background bacteria, and there is no relevant research on using the microbial model to predict product safety.

In this study, the growth kinetic models of *Salmonella* with the appearance of dominant background bacteria in pork at different temperatures were established, to be used for the prediction of the shelf life of pork, which would provide scientific guidance for pork transportation, storage, and safety monitoring.

## 2. Results

### 2.1. Optimal Primary Growth Kinetic Model of Salmonella and Two Background Bacteria in Fresh Pork

According to the experimental observation data of *Salmonella* Derby (*S.* Derby), *Pseudomonas aeruginosa* (*P. aeruginosa*), and *E. coli* in fresh pork during storage at 4~37 °C, the primary growth kinetic models of these three bacteria at different temperatures were obtained by data fitting using modified Gompertz, Baranyi, and Logistic equations, respectively, and the coefficient of determination *R*^2^ of each fitting growth curve is listed in Table 1. Since the bacterial quantity did not change significantly compared with the initial quantity after storage at 4 °C for 10 d, the curve could not be fitted. The Baranyi model could best fit the growth curves of *S.* Derby and *P. aeruginosa* in fresh pork at 37, 30, 22, 16, and 10 °C (Figure 1), and all the values of the coefficient of determination (*R*^2^) at different temperatures were higher than the modified Gompertz and Logistic models. The Huang model, with higher *R*^2^ values, was better than the other two when fitting the growth curves of *E. coli* in pork at different temperatures (Figure 1). Therefore, the Baranyi model was chosen as the optimal primary growth model for *S.* Derby and *P. aeruginosa*, and the Huang model was chosen as the optimal model for *E. coli* in this study. The kinetic parameters, *N_max_*, *μ_max_*, and *Lag*, of each optimal bacterial primary growth model under five temperatures, as well as the observed initial bacterial quantity (*N*_0_), are listed in Table 2. With the increase in temperature, the maximum growth rates of the three bacteria in pork all increased, while the lag period decreased.

### 2.2. Secondary Growth Model of S. Derby, P. aeruginosa, and E. coli in Fresh Pork

According to the primary model, the parameters *μ_max_* and *Lag* of *S.* Derby, *P. aeruginosa*, and *E. coli* at different temperatures were separately fitted using the square root model to reflect the relationship between temperature with maximum growth rate and lag phase (Figure 2). The related expression equation of each secondary growth model is also shown in Figure 2. From the results, it was found that the square root of *μ_max_* and *1/Lag* of each bacterium presented a nice linear relationship with temperature, with most *R*^2^ values above 0.98, indicating higher model fitting degrees.

### 2.3. Validation of the Bacterial Growth Models

Bias factor *B_f_* and accuracy factor *A_f_* are often used as effective tools to evaluate model reliability [19]. A model is considered better and is acceptable when *B_f_* is in the range of 0.90~1.05 and *A_f_* is in the range of 1.01~1.15. The closer *A_f_* and *B_f_* are to 1, the more reliable the model [20]. The smaller the *MSE* value, the smaller the variability of the data and the higher the accuracy of the model. Therefore, *B_f_*, *A_f_*, and *MSE* were used to evaluate the constructed secondary model. From the data in Table 3, it was found that the *B_f_* and *A_f_* of the secondary model fitted by the temperature with *μ_max_* and *Lag* of the three bacteria are all close to 1, and all the values of *MSE* are very small, which indicates that the square root model can be used to predict the relationship between temperature and *μ_max_* and *Lag* with high reliability.

### 2.4. Pork Safety Prediction by Applying the Constructed Growth Models

The simulated mixed-contamination pork stored at different temperatures was dynamically observed and scored according to the sensory indicators, and the earliest time to reach the corresponding level standard was recorded. It was found that the higher the temperature, the shorter the time for pork from fresh to spoilage (Table 4). At the same time, the time required for *S.* Derby to reach the unsafe bacterial dose was calculated using the derived shelf life model, and was compared with the sensory change time. The calculated time data at each temperature were found to be relatively close to the elapsed time when the sensory score reached sub-fresh, and earlier than the elapsed time when the sensory score reached nearly fresh. This indicated that the content of pathogenic bacteria in pork might already exceed the safe dose even if the color and smell of the pork are basically normal. Therefore, determining the shelf life of food according to the predicted microbiology could more scientifically and accurately ensure the consumption safety of the product.

## 3. Discussion

The contamination of pathogenic bacteria and spoilage bacteria affects the quality and safety of pork products [1]. Predictive microbiology provides a more scientific approach to product risk and shelf-life prediction. There are few reports on growth predictive models of multiple bacteria in fresh pork. In this experiment, the growth curves of three bacteria in fresh pork were studied at a common temperature (4~37 °C) during storage and transportation, and the optimal models were selected to simulate their growth, which can dynamically predict the spoilage extent of pork. This study could provide guidance to ensure hygiene throughout the pork industrial chain and to determine pork shelf life more scientifically.

There are many studies and applications of predictive microbiology in food. The research on fresh livestock and poultry products is mostly concentrated on bacteria in poultry or cattle products, and there are also predictive microbiology studies on mixed bacterial contamination [8,17,21,22,23,24]. However, current predictive microbial studies on pork mainly target a single pathogen [25,26]. Given that the microorganisms in swine are exposed during the whole chain from breeding and slaughtering to sale and consumption cannot be single, it is more meaningful to study mixed predictive microbiology in pork. Here, *S.* Derby, *E. coli*, and *P. aeruginosa* were selected to analogously contaminate pork in a laboratory, and the predicted growth kinetics of the three bacteria at 4~37 °C were studied. Although the constructed model was not an integrated competitive growth model, the growth process of the three bacteria in pork was in a competitive mode. Therefore, the model presents a growth curve under a competitive mode.

Although different nonlinear equations, such as modified Gompertz, Baranyi, and Huang models, were used to predict microbial growth, it was not clear which model was inherently better than the other [27]. The determination of the optimal model is inseparable from the culture medium, experimental conditions, bacterial species, and other factors. For a complex meat system, the optimal model, rather than a single model, should be selected to predict the bacterial growth under different temperatures [28]. In this study, Baranyi models could better simulate the growth of both *S.* Derby and *P. aeruginosa* at different temperatures, which is consistent with previous research findings, while the optimal model of *E. coli* is the Huang model. There are recent studies that use the Huang model to fit the growth of *E. coli* in beef and chicken [29,30], but there is no study using the Huang model to fit the growth of *E. coli* in fresh pork. Therefore, the results of this study broaden the application of this model in the field of microbiology to a certain extent. Although a mixed inoculation of three bacterial strains were applied in this study, there was no clear competitive inhibition among strains, and all three strains showed an S-shaped growth curve, which may be related to the relatively similar amount of initial inoculum.

According to the growth kinetic parameters obtained by the optimal primary models of the three bacteria in pork, with the increase in temperature, the *μ_max_* value increased, while the *Lag* value decreased, indicating that temperature has a significant effect on bacterial growth in pork. Compared with the predictive growth models of related bacteria in pork in previous studies [31,32], the model obtained through mixed inoculation in this study showed a higher *μ_max_* value and lower *Lag* value, which may be due to the quorum sensing between the inoculated mixed strains. Quorum sensing can cause bacteria to mutually promote growth and shorten the time taken to adapt to external conditions [33,34]. In fact, differences in inoculation method and amount, bacterial characteristics, meat types, etc. during experiments would lead to differences in *μ_max_* and *Lag*.

In addition, it is worth noting that in this study, under the condition of 10 °C, pork could be stored for 192 h and was still nearly fresh. The reason should be that the initial bacterial inoculation amount was quite low (100~300 CFU), and the microbial lag period in pork was longer; therefore, the quorum sensing could not be formed quickly, which meant that the bacteria could not reproduce quickly. In actual production, the total number of bacterial colonies in pork sometimes exceeds 1000 CFU; therefore, the pork may deteriorate within 1 week of storage under refrigeration (4~10 °C).

In summary, the predictive growth kinetic models of *Salmonella*, *Pseudomonas*, and *E. coli* inoculated simultaneously in fresh pork at different temperatures were each constructed and validated, and the shelf life of pork for safe consumption predicted based on the predicted growth model was more secure when compared with the conventional sensory evaluation indicators of pork.

## 4. Materials and Methods

### 4.1. Bacterial Cultures and Inoculum Preparation

*S.* Derby is the most dominant pathogen in pork in Shandong province, *P. aeruginosa* is one of the dominant spoilage bacteria [35], and *E. coli* is one of the most common background bacteria. Therefore, we chose these three bacteria to study their predictive growth models in pork at different temperatures. *S.* Derby, *P. aeruginosa*, and *E. coli* were inoculated onto LB agar (Beijing Land Bridge Technology, Beijing, China) and cultured at 37 °C for 16 h. One typical single colony of each strain was selected and transferred into 5 mL of LB liquid medium (Beijing Land Bridge Technology, Beijing, China), and cultured at 37 °C with shaking (200 r/min) for 16 h. Then, three types of bacterial culture were centrifuged at 3000 r/min at 4 °C for 10 min, the supernatant was discarded, sterile saline was added to resuspend the bacteria, and then gradient dilution was carried out to about 10^2^ CFU/mL of final concentration. An equal volume of each bacterial solution was mixed to serve as the bacterial inoculum.

### 4.2. Preparation of Pork Sample and Inoculation

Fresh pork purchased from a local supermarket was rinsed with sterile water, the surface was wiped with alcohol cotton in sterile biological safety cabinet, and then the whole pork was rinsed with sterile water again. The pork’s surface layer was removed using a sterile scalpel, the pork was divided into small pieces (about 10 g/piece), and then the pork pieces were placed on aseptic trays to be sterilized again with ultraviolet rays for 25–30 min. The sterile pork piece samples were suspended in the bacterial mixed suspension for 15 s for inoculation to yield an initial inoculum of about 10 CFU/g.

### 4.3. Growth Study and Bacterial Counting

Each piece of inoculated pork sample was placed in one sterile homogenizing bag, and cultured in a constant temperature incubator at 4, 10, 16, 22, 30, and 37 °C, respectively. Before culture, one simulated contaminated pork piece of each temperature was randomly selected to carry out bacterial counting to determine the initial amount of contamination of these three bacteria. At the same time, bacterial counting was conducted for the control pieces without inoculation to ensure no background bacterial contamination. No background bacteria were detected only in the control pieces, and the experiments were continued.

Under each temperature, one sample was taken out in a timely manner and added to 90 mL of sterile normal saline, beaten at a speed of 7 times per second for 2 min. The time points at 37 °C were set as 0, 0.25, 0.5, 1, 1.5, 2, 3, 4, 5, 6, 7, 8, 9, 11, and 13 h; the time points at 30 °C were set as 0, 0.5, 1, 2, 3, 4, 5, 6, 7, 8, 9, 10, 12, 14, 16, and 20 h; the time points at 22 °C were set as 0, 1.5, 3, 6, 9, 13, 17, 21, 24, 27, 31, 35, 40, 44, and 48 h; the time points at 16 °C were set as 0, 2.5, 5, 10, 15, 20, 27, 32, 44, 50, 55, and 60 h; the time points at 10 °C were set as 0, 24, 48, 72, 96, 120, 14, 168, 192, 216, and 240 h. Then, the liquid portion of each sample was properly diluted, and 100 μL of aliquots of suitable dilution gradients was plated on *Salmonella* chromogenic medium (CHROMagar™, Paris, France), MacConkey medium (Beijing Land Bridge Technology, Beijing, China), as well as CFC Pseudomonas selective medium (Beijing Land Bridge Technology, Beijing, China) one-by-one. The plates were incubated at 37 °C for 24 h. The bacterial colonies with corresponding colors on each kind of agar plate were counted and converted to log CFU/g of sample. Each experiment was carried out in triplicate.

### 4.4. Sensory Evaluation of Pork Freshness

The freshness of pork was evaluated based on its color, texture, and smell [36]. The specific evaluation grades and criteria of sensory indicators are shown in Table 5.

### 4.5. Establishment of the Primary Bacterial Growth Model

The growth data of the three bacteria obtained in the experiment were fitted by the modified Gompertz model in Origin 2019 software (Equation (1)) [37], by the Huang model in IPMP2013 software (Equations (2) and (3)), and by the Baranyi model in the online Combase database (Equations (4) and (5)) [14,38]. The coefficient of determination (*R*^2^) of the three different models was compared, and the best model was selected.

Modified Gompertz model:*N_t_* = *N*_0_ + (*N_max_* − *N*_0_) × exp{−exp [2.718*μ_max_*/(*N_max_* − *N*_0_) × (*Lag* − *t*)] + 1} (1)
where *t* is the bacterial growth time (h); *N_t_* is the corresponding bacterial concentration at *t* (log CFU/g); *N_max_* and *N*_0_ are the maximum and initial bacterial concentration (log CFU/g), respectively; *μ_max_* is the microbial maximum growth rate (h^−1^); and *Lag* is the bacterial growth delay time (h).

Huang model:*N_t_* = *N*_0_ + *N_max_* − ln[e*^N^*_0_ + (e*^Nmax^* − e*^N^*_0_)e^−*μmax Bt*^] (2)
(3)$$B_{t}=t+\frac{1}{\alpha }\ln\frac{1+e^{-\alpha \left(t-Lag\right)}}{1+e^{\alpha Lag}}$$
where *t* is the bacterial growth time (h); *N_t_* is the corresponding bacterial concentration at *t* (log CFU/g); *N_max_* and *N*_0_ are the maximum and initial bacterial concentration (log CFU/g), respectively; *μ_max_* is the microbial maximum growth rate (h^−1^); *Lag* is the bacterial growth delay time (h); and α is a constant of 4, which defines the transition from delayed lag phase to logarithmic phase during bacterial growth.

Baranyi model:(4)$$N(t)=N_{0}+\mu_{\max}A(t)-\ln\;[1+\frac{e^{\mu_{max}A\left(t\right)}-1}{e^{\left(N_{max}-N_{0}\right)}}]$$
(5)$$A(t)=t+\frac{1}{\mu_{max}}\;\ln(e^{-}\;{}^{t\mu \max}+e^{-}\;{}^{\mathrm{Lag}\mu \max}-e^{-}\;{}^{t\mu \max-}\;{}^{\mathrm{Lag}\mu \max})$$

In the above five equations, *t* is the bacterial growth time (h); *N_t_* is the corresponding bacterial concentration at *t* (log CFU/g); *N_max_* and *N*_0_ are the maximum and initial bacterial concentration (log CFU/g), respectively; *μ_max_* is the microbial maximum growth rate (h^−1^); *Lag* is the bacterial growth delay time (h); and α in Equation (3) is a constant of 4, which defines the transition from delayed lag phase to logarithmic phase during bacterial growth.

### 4.6. Establishment of the Secondary Bacterial Growth Model

The Ratkowsky square root model can well predict the growth of microorganisms under single-factor condition, which is often used to describe the relationship between temperature, maximum growth rate, and lag period [39,40]. The relationship between temperature and *Lag* and *μ_max_* was fitted using Origin software. The expressions’ square root equation of maximum growth rate and lag period are as in Equations (6) and (7):(6)$$\sqrt{1/Lag}=b_{\mathrm{Lag}}\;\times \;(T-T_{\min})$$
(7)$$\sqrt{\mu_{max}}=b_{\mu \max}\;\times \;(T-T_{\min})$$
where *T* is the incubation temperature (°C); *T_min_* is the minimum growth temperature, namely, the temperature at which the microorganism has no metabolic activity, with a maximum growth rate of 0; and b is a constant of the equation.

### 4.7. Validation of the Bacterial Growth Model

The primary bacterial growth models were verified using decisive coefficient (*R*^2^), which was directly obtained from data fitting results by the modified Gompertz model in Origin 2019, the Huang model in IPMP2013, and the Baranyi model in the online Combase database. The secondary bacterial growth models were verified using *R*^2^, mean square error (*MSE*, Equation (8)), bias factor (*B_f_*, Equation (9)), and accuracy factor (*A_f_*, Equation (10)) [19,20,41]:(8)$$\mathrm{MSE}=\frac{{\sum (observed-predicted)}^{2}}{n}$$
(9)$$B_{f}=10^{\frac{\sum log\left(predicted/observed\right)}{n}}$$
(10)$$A_{f}\;=10^{\frac{\sum |log\;(predicted/observed)|}{n}}$$
where *predicted* represents the predicted value of *μ_max_* and *Lag*, which was calculated from the equation of the secondary model; *observed* represents the observed value of *μ_max_* and *Lag*, which was obtained from the primary model; and n represents the number of observations (here, n = 5).

### 4.8. Shelf-Life Prediction of Fresh Pork

The safety of fresh pork was mainly considered in terms of shelf-life prediction in this study. Through the constructed bacterial primary and secondary growth models, the shelf life (*SL* (h)) of fresh pork was predicted based on the *S.* Derby growth time required from the initial bacterial quantity (*N*_0_) to the minimum pathogenic dose (*N_s_*). The shelf-life prediction equation was derived from the simplified Baranyi model [42] (Equation (11)).
(11)$$\mathrm{SL}=\frac{\;Ln\;\left[e^{\left(Ns-N0+Lag\mu max\right)}+e{\;}^{\left(Lag\mu max\right)}+1\right]}{\mu max}$$

## Figures and Tables

**Figure 1 molecules-27-07673-f001:**
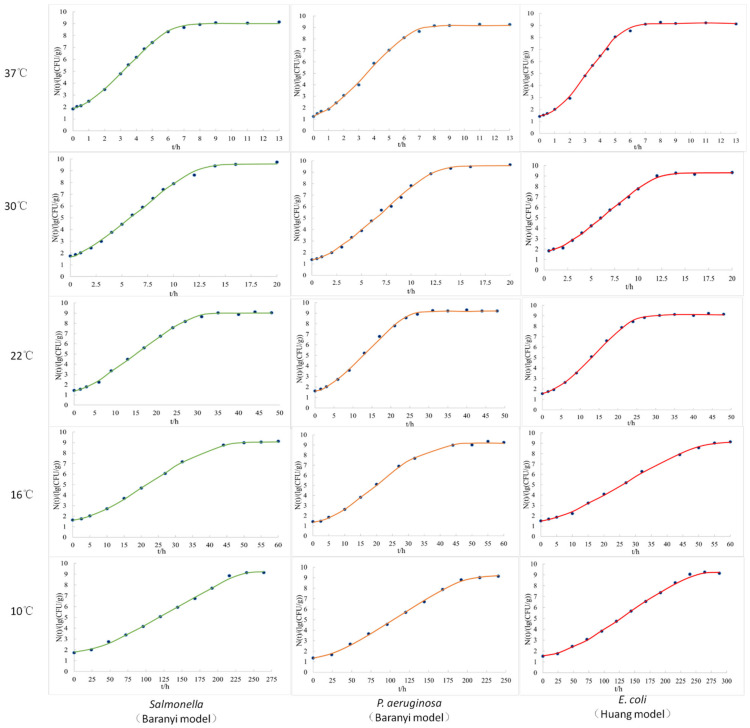
Growth curves of *S.* Derby, *P. aeruginosa*, and *E. coli* in fresh pork at different temperatures fitted by the corresponding optimal predictive model.

**Figure 2 molecules-27-07673-f002:**
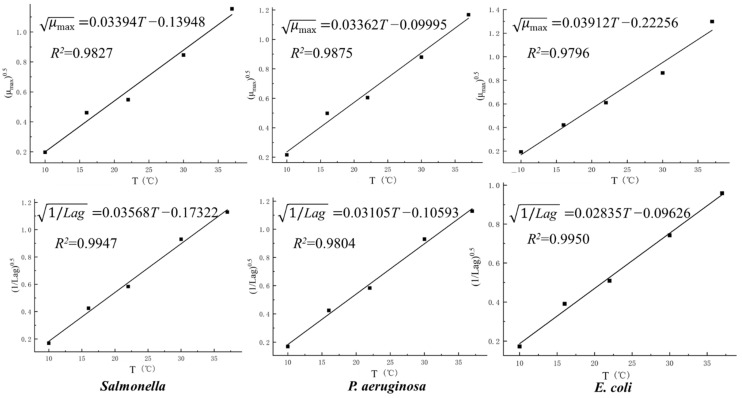
Secondary square root models of *S.* Derby, *P. aeruginosa*, and *E. coli* in pork.

**Table 1 molecules-27-07673-t001:** Decisive coefficient (*R*^2^) of fitting curve of each model.

Bacteria	Model	*R* ^2^				
		37 °C	30 °C	22 °C	16 °C	10 °C
*S.* Derby	Gompertz	0.9871	0.9930	0.9922	0.9924	0.9903
	Baranyi	0.9993	0.9982	0.9990	1.0000	0.9969
	Huang	0.9985	0.9976	0.9974	0.9989	0.9950
*P. aeruginosa*	Gompertz	0.9891	0.9845	0.9852	0.9921	0.9915
	Baranyi	0.9988	0.9973	0.9990	0.9989	0.9992
	Huang	0.9946	0.9910	0.9983	0.9971	0.9962
*E. coli*	Gompertz	0.9830	0.9823	0.9872	0.996	0.9900
	Baranyi	0.9980	0.9973	0.9987	0.9976	0.9970
	Huang	0.9983	0.9986	0.9991	0.9989	0.9972

**Table 2 molecules-27-07673-t002:** Kinetic parameters of each bacterial optimal primary growth model in pork at different temperatures.

Bacteria	Temperature (°C)	Growth Kinetic Parameters			
		*N*_0_ (logCFU/g)	*Lag* (h)	*μ_max_* (h^−1^)	*N_max_* (logCFU/g)
*S.* Derby	37	1.8641	0.7844	1.3325	9.0103
	30	1.6846	1.1565	0.7157	9.5800
	22	1.3657	2.9393	0.3000	9.0076
	16	1.6099	5.5342	0.2127	9.0649
	10	1.7850	34.8147	0.0387	9.4430
*P. aeruginosa*	37	1.3220	0.8450	1.3627	9.1980
	30	1.3412	1.6851	0.7738	9.5601
	22	1.6104	3.4016	0.3660	9.2053
	16	1.3383	5.1190	0.2483	9.1722
	10	1.3311	25.3870	0.0467	9.2507
*E. coli*	37	1.5676	1.0880	1.6853	9.1748
	30	1.8894	1.8135	0.7450	9.3809
	22	1.7443	3.8602	0.3725	9.1267
	16	1.6820	6.5438	0.1774	9.2960
	10	1.6392	33.6302	0.0372	9.5750

**Table 3 molecules-27-07673-t003:** Various statistical characteristics of the secondary growth models.

Parameter	Bacteria	*B_f_*	*A_f_*	*MSE*
*μ_max_*	*S.* Derby	0.9981	1.0670	0.0019
	*P. aeruginosa*	1.0056	1.0675	0.0013
	*E. coli*	0.9812	1.0783	0.0029
*Lag*	*S.* Derby	1.0080	1.0503	0.0014
	*P. aeruginosa*	0.9993	1.0675	0.0018
	*E. coli*	1.0054	1.0465	0.0013

**Table 4 molecules-27-07673-t004:** Sensory changes in simulated contaminated pork at different temperatures and prediction of shelf life based on *S.* Derby amount.

Sensory Quality	Grade	Store Time (h)				
		37 °C	30 °C	22 °C	16 °C	10 °C
Fresh	5	0	0	0	0	0
Sub-fresh	4	4	7	14	23	93
Nearly fresh	3	8	12	27	44	192
Not fresh	2	11	15	31	60	264
Spoilage	1	17	24	44	65	288
*S.* Derby (10^5^ CFU/g)	-	3.18	5.85	15.17	21.67	119.16

Note: After 14 days at 4 °C, the indicators did not change and were still fresh products, which may be due to the weak bacterial activity.

**Table 5 molecules-27-07673-t005:** Sensory evaluation criteria.

Grade	Sensory Traits	Quality
5	Bright red, milky-white fat, shiny, quick recovery from acupressure, slightly dry appearance, non-sticky, normal odor	Fresh
4	Bright red, slightly darkened fat, slightly reduced luster, recovered after acupressure, slightly wet, slightly sticky, normal odor	Sub-fresh
3	Color appears slightly darker, decreased luster, recovery of acupressure becomes slow, slightly wet, slightly sticky, no clear odor change	Nearly fresh
2	Dark red, slightly greenish fat, very slow recovery of acupressure, moist, sticky, with a peculiar smell	Not fresh
1	Dark red, green surface, no recovery of acupressure, a large amount of mucus, very sticky, with a pungent smell	Spoilage

## Data Availability

Not applicable.

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
