# Peer review of "Establishment and Application of a Predictive Growth Kinetic Model of Salmonella with the Appearance of Two Other Dominant Background Bacteria in Fresh Pork"

_molecules, 2022, doi:10.3390/molecules27227673_

Round 1

Reviewer 1 Report

Authors presented a paper on using kinetic models to predict Salmonella, Pseudomonas aeruginosa, and E.coli bacteria growth in pork. There are several issues with the manuscript. 

Firstly, the discussion section is empty. Within Result section, there is a paragraph titled 3.3, is that due to incorrect formatting? In any case, the omission of writing the whole discussion section must be addressed. 

Secondly, it is very interesting idea to model 3 bacteria at the same time. But the traditional kinetic modelling used in the study did not take into account the possible interactions with each other and other factors apart from temperature. There are new methodological framework to use machine learning (ML) to achieve this goal. And methods were introduced integrating ML, growth model, and/or survival database to predict the bacterial growth. 

Author Response

Q1: Firstly, the discussion section is empty. Within Result section, there is a paragraph titled 3.3, is that due to incorrect formatting? In any case, the omission of writing the whole discussion section must be addressed. 

Answer: Thank you for pointing this out in a timely manner, the discussion section was missed in the final layout of the article and has now been added.  (L277-510)

Q2: Secondly, it is very interesting idea to model 3 bacteria at the same time. But the traditional kinetic modelling used in the study did not take into account the possible interactions with each other and other factors apart from temperature. There is new methodological framework to use machine learning (ML) to achieve this goal. And methods were introduced integrating ML, growth model, and/or survival database to predict the bacterial growth. 

Answer: Thank you for raising a very good suggestion. The current study was indeed just focus on the effect of temperature on Salmonella growth in pork in the presence of two background bacteria. We will incorporate the machine learning method that you mentioned in our future research.

Reviewer 2 Report

Dears, the manuscript sound interesting and deserve to be published. My only consideration is: i realizes thar you use statist measures in the  growth models, but I could not found the methods in the M&M. Please provided it. 

Author Response

Thank you for pointing this out in a timely manner. We added the statistic measures of bacterial growth model analysis, especially the calculation method of validation parameters, in 4.7 section of the M&M.

Reviewer 3 Report

Dear Authors,

The present manuscript need to be improved. The English of the article is not understandable; there are too long sentences and too many verbs have been used in the same sentence.

There is no discussion part throughout the article.

The genus and species names of the microorganisms should be written correctly according to the scientific paper writing rules.

The references of the manuscript should be re-arranged according to the "Instruction for Authors" manual of the Journal. According to the "Instruction for Authors" of the "Molecules", references must be numbered in order of appearance in the text (including table captions and figure legends) and listed individually at the end of the manuscript. Please check the References throughout the manuscrpit.

Author Response

Q1: Dear Authors, The present manuscript need to be improved. The English of the article is not understandable; there are too long sentences and too many verbs have been used in the same sentence.

Answer: We apologize for the poor language in our manuscript. We invited native English speakers from the links provided by the editing office to polish the English language of the whole manuscript, and we really hope the language is now substantially better. The yellow mark is modified by the author, and the annotation is modified by English Editing.

Q2: There is no discussion part throughout the article.

Answer: Thank you for pointing this out in a timely manner, the discussion section was missed in the final layout of the article and has now been added. See the appendix for details. (L277-510)

Q3: The genus and species names of the microorganisms should be written correctly according to the scientific paper writing rules.

Answer:Yes, it's our negligence. We've revised all the writing of genus and species names of the microorganisms in the revised manuscript.

Q4: The references of the manuscript should be re-arranged according to the "Instruction for Authors" manual of the Journal. According to the "Instruction for Authors" of the "Molecules", references must be numbered in order of appearance in the text (including table captions and figure legends) and listed individually at the end of the manuscript. Please check the References throughout the manuscript.

Answer: All references have been revised as required in the revised manuscript. 

Round 2

Reviewer 1 Report

The authors have made some necessary improvement to the manuscript. The discussion section in particular received due modification. Overall, the manuscript presented an interesting work to the community. Some minor issues may have to be addressed. For example, the language can still be improved by using more concise and simple sentences. Some sentences can use another polish. 

Reviewer 3 Report

Required corrections have been done by the authors.